# Accelerated Aging and Age-Related Diseases (CVD and Neurological) Due to Air Pollution and Traffic Noise Exposure

**DOI:** 10.3390/ijms22052419

**Published:** 2021-02-28

**Authors:** Omar Hahad, Katie Frenis, Marin Kuntic, Andreas Daiber, Thomas Münzel

**Affiliations:** 1Department of Cardiology, University Medical Center Mainz, Johannes Gutenberg University, 55131 Mainz, Germany; omar.hahad@unimedizin-mainz.de (O.H.); katiefrenis@gmail.com (K.F.); marin.kuntic93@gmail.com (M.K.); 2German Center for Cardiovascular Research (DZHK), Partner Site Rhine-Main, 55131 Mainz, Germany

**Keywords:** aging, air pollution, traffic noise exposure, oxidative stress, inflammation, cardiovascular disease, neurological disease

## Abstract

The World Health Organization estimates that only approximately 25% of diversity in longevity is explained by genetic factors, while the other 75% is largely determined by interactions with the physical and social environments. Indeed, aging is a multifactorial process that is influenced by a range of environmental, sociodemographic, and biopsychosocial factors, all of which might act in concert to determine the process of aging. The global average life expectancy increased fundamentally over the past century, toward an aging population, correlating with the development and onset of age-related diseases, mainly from cardiovascular and neurological nature. Therefore, the identification of determinants of healthy and unhealthy aging is a major goal to lower the burden and socioeconomic costs of age-related diseases. The role of environmental factors (such as air pollution and noise exposure) as crucial determinants of the aging process are being increasingly recognized. Here, we critically review recent findings concerning the pathomechanisms underlying the aging process and their correlates in cardiovascular and neurological disease, centered on oxidative stress and inflammation, as well as the influence of prominent environmental pollutants, namely air pollution and traffic noise exposure, which is suggested to accelerate the aging process. Insight into these types of relationships and appropriate preventive strategies are urgently needed to promote healthy aging.

## 1. Introduction

The dramatic improvement in life expectancy over the past century led to an unprecedented demographic shift toward an aging population; the proportion of the population over 65 is higher than ever before. As the population boom of the 20th century ages, age-related diseases have come to the forefront as emergent health concerns [1]. In contrast to maternal, infectious diseases that were widely prevalent and a primary health concern of the early 20th century, age-related diseases are often chronic and require continual treatment over an extended period of time, thus correlating increased lifespan with chronic disease onset and elevated expense burden. Aging is a multifactorial dynamic process that is influenced by a variety of external and internal variables, including environmental, demographic, and biopsychosocial factors, to determine the development and progression of age-related diseases, rather than being a solely static intrinsic process of cellular alterations (Figure 1).

The aging population is particularly susceptible to cardiovascular disease (CVD), demonstrating the leading cause of death in populations aged over 65 years (Figure 2), and creating an urgent need for research in the field. Compounding the rise in CVD prevalence, as age advances, there is also a rise in complications and comorbidities of CVD [4,5,6]. This phenomenon is partly due to the “silent” nature of CVD pathophysiological development, but also due to vascular aging, which represents all changes in the vessels over time that exacerbate disease development [7]. Specifically, aged vessels have an impaired endothelium and other constitutional changes, which make them more prone to atherosclerotic lesions, vascular injury, and calcification, alongside blunted angiogenesis [8]. With age, the endothelium displays decreased responsiveness that manifests endothelial dysfunction in elderly people [9,10] and corresponds well with the demographic data that associates age, CVD incidence, and CVD comorbidities [11]. Endothelial impairment leading to endothelial dysfunction is also accompanied by smooth muscle changes, since arterial stiffening is also observed in aged vessels. Both mechanistic aspects correlate with future cardiovascular events in humans [12].

The same holds true for the aging brain (Figure 2). The incidence of stroke shows a dramatic increase in the elderly [17], and cognitive impairment clearly progresses with age and represents an accepted early diagnostic parameter for later dementia and neurodegeneration [18]. When looking at the risk factors of dementia in detail, it becomes clear that there is a large overlap with cardiovascular risk factors [19]. Mounting evidence indicates that the aging process is fundamentally driven by environmental exposures, and interestingly, age-related pathomechanisms were also observed in the context of predominant environmental pollutants, such as air pollution [20,21] and (traffic) noise exposure [22,23], with growing evidence suggesting that these pollutants might cause or accelerate age-related diseases. In this review, we discuss the pathomechanisms underlying the aging process and their correlates in CVD and neurological disease development, with a main focus on oxidative stress pathways and inflammation. Furthermore, we critically review emerging findings concerning environmental factors, namely air pollution and noise exposure, which affect and accelerate the aging process.

## 2. Impact of Aging on Inflammation, Adverse Redox Signaling, Endothelial Dysfunction, and CVD

Endothelial dysfunction is an important indicator of subclinical CVD and serves as an early predictor of developing atherosclerosis, hypertension, and future cardiovascular events. There are two critical mechanisms through which endothelial dysfunction influences pathogenesis within the context of vascular aging. First, it promotes vasoconstriction, thrombocyte activation, leukocyte infiltration, and smooth muscle cell proliferation in the vessel wall; all of which precede cardiovascular events. The second is due to impaired endothelial signaling in all vessels; age-dependent endothelial dysfunction is found in both macrovessels and resistance vessels (for review see [24]), and thereby can impact a wide variety of disease states.

Three interdependent players are known to trigger endothelial dysfunction—inflammation, oxidative stress, and impaired nitric oxide (^•^NO) signaling [25]. Endothelial oxidative stress, an important trigger of endothelial dysfunction, is associated with age-related diseases other than CVD, including erectile dysfunction, renal dysfunction, Alzheimer’s disease, or retinopathy [26,27,28,29]. The studies of Mayhan et al. highlight these findings, demonstrating that cerebral arterioles show diminished eNOS-dependent reactivity, which positively correlated with increased oxidative stress in aged rats [30]. These findings were echoed in studies in other vessels and conditions, showing that endothelial dysfunction and oxidative stress are present in aging retinal vessels [31], and are a contributing factor in Alzheimer’s and Parkinson’s diseases, through several mechanisms [32,33]. Lastly, oxidative stress in combination with vascular inflammation and impaired ^•^NO signaling were identified as key players in age-related endothelial dysfunction by our group and many others (for review see [34,35]). As aged vessels show strong associations with oxidative imbalances, inflammatory increases, and negative effects on ^•^NO signaling, aging is implicated as an independent risk factor for CVD [36,37].

In many ways, a reciprocal and interdependent relationship exists between oxidative stress deriving from mitochondrial or enzymatic sources, endothelial dysfunction, and hypertension, diabetes, and atherosclerosis. It is unsurprising then, that oxidative stress [38,39], endothelial dysfunction [36], and the aforementioned CVDs [40,41] all see an increased incidence with advancing age, as they often occur in parallel, exert some influence on each other, and also have associations with low-level inflammation. Accordingly, age-dependent changes in the composition and function of high-density lipoproteins (HDL) were reported [42], which further underlined the contributing mechanisms of the risk factors previously discussed, since HDL inhibit inflammation, have antioxidant properties [43], and inversely correlate with coronary disease risk [43]. The degradation of HDL quality over time negatively impacts endothelial function, a critical factor in the initiation and development of atherosclerosis, potentially indicating HDL as a target for therapeutic intervention of age-related CVD [44].

Hypertension, the predominant risk factor for atherosclerosis and other CVD, potentiates the causative elements behind endothelial dysfunction, making effective treatment of hypertension an important route for the prevention of age-related CVD. To this end, pregnant spontaneously hypertensive rats were treated with nitrovasodilator pentaerythrityl tetranitrate, which demonstrated blood pressure lowering effects that were inherited by offspring. It was found that enhanced histone 3 lysine 27 acetylation and histone 3 lysine 4 trimethylation (epigenetic markers usually associated with transcriptional activation) promoted the transcriptional activation of cardioprotective genes like *eNOS*, *SOD2*, *GPx-1*, and *HO-1*, which explained the observed heritable effects [45]. Drugs with epigenetic effects, like pentaerithrityl tetranitrate, could conceivably be used to extend the number of healthy years, and perhaps stave off the effects of cardiovascular aging. A third possible therapeutic strategy would utilize mitochondria-targeted antioxidants to mitigate the “side effects” of the aging process. Treatment with dietary vitamins equating to unspecific antioxidant treatment was not found to be effective in preventing vascular aging. However, specifically targeting mitochondrial ROS could represent a possible strategy to alleviate, at least in part, age-related endothelial dysfunction. Along this line, age-related endothelial dysfunction was alleviated by administration of mito-quinone in mice [46]. Some risk factors for CVD could be changed by lifestyle alterations, such as smoking and diet [47], but aging is a factor that is not preventable, and so must be tackled in a bottom-up approach. 

As previously mentioned, low-level inflammation is commonly found in aged individuals. One study found that plasma levels of important inflammatory markers, including soluble vascular adhesion molecule 1(sVCAM-1), interleukin 6 (IL-6), and monocyte chemoattractant protein 1 (MCP-1) positively correlated with age, even where there was no underlying CVD or risk factors present [48]. Another study found positive correlations between age and levels of circulating IL-6, IL-1 receptor antagonist (IL-1ra), IL-18, C-reactive protein (CRP), and fibrinogen, in both men and women, most persisting after correction for other risk factors. Increases of soluble IL-6 receptor (sIL-6r) occurred with greater age, but this effect was only noted in men [49]. A meta-analysis spanning 32 cross-sectional studies and over 23,000 subjects revealed associations between serum CRP and IL-6 levels and the onset or presence of frailty and pre-frailty, a phenotype that encompassed unintentional weight loss, exhaustion, weakness, slow walking speed, or low physical capability [50]. The hazard ratio for serum CRP levels and incidence of frailty was 1.06 (95% confidence interval [CI] 0.78–1.44), alongside a hazard ratio of 1.19 (95% CI 0.87–1.62) for IL-6, after adjustment for 9 potential confounders [50], illustrating a correlation between the presence of inflammation and age-related ailments. 

Air and noise pollution are novel cardiovascular risk factors whose mechanisms are still being investigated, but have so far shown similar molecular signatures [51,52,53] to one another, as well as to classical risk factors like hypertension, hypercholesterolemia, or hyperglycemia [54,55,56]. Foremost amongst these signatures appears to be oxidative stress and inflammation, which both mediate the detrimental effects following exposure to noise and air pollution [57]. Despite some clarity as to the mechanisms, it is not fully understood how crosstalk between stress response pathways, redox signaling, and the immune system coordinate to cause cardiovascular damage in response to these novel risk factors.

## 3. Impact of Aging on Inflammation, Adverse Redox Signaling, Neuronal Degeneration, and Neurological Disease

Since CVDs and neurological diseases have a substantial overlap in risk factors and pathophysiological pathways, it is important to highlight the aforementioned mechanisms of action in the context of aging and neurological disease. In general, functional and structural deterioration of the aging brain is a cumulative process that starts with subclinical alterations at the molecular level. These changes include accumulation of mutations, telomere attrition, and epigenetic alterations, resulting in genomic instability and thus priming for neuronal damage and loss, reduced neurotransmitter levels, enhanced neuroinflammation, increased susceptibility to cerebral ROS, and decreased cerebral vascular compliance. All of these adverse processes are associated with increased risk of age-related neurological diseases, such as stroke, epilepsy, Parkinson’s disease, and dementia/cognitive decline [58]. Immunosenescence and inflammaging, as the most recognized effects of aging [59], might promote neuroinflammatory processes along with cerebral oxidative stress, via altered microglia activation (immune cells of the brain), which are central to neurotoxicity through the release of neurotoxic cytokines, such as TNFα, IL-1β, and INF-γ, as well as different ROS such as ONOO^−^ and O_2_^•−^ [60,61]. Microglial dysregulation represents a hallmark of various neurological complications, and adverse redox regulation of and by microglia plays a crucial role in these processes [62,63,64]. Neuroinflammation and cerebral oxidative stress might act together to increase neuronal damage/loss and amyloid deposition, as well as to decrease cerebral ^•^NO bioavailability via NOX-2 activation and uncoupling of neuronal ^•^NO synthase (nNOS), leading to cerebral vascular endothelial dysfunction and ultimately contributing to increased risk of stroke, epilepsy, Parkinson’s disease, and dementia/cognitive decline in the elderly [21]. 

From an epidemiological point of view, the accelerated aging of the population and the correspondent increase in the elderly would affect the number of patients with neurological diseases, as recently demonstrated by results of the Dijon Stroke Registry. In this study, an increase of 55% in the total annual number of stroke cases by 2030 was calculated, largely driven by increased prevalence in the group of elderly people (65% in people ≥ 75 years vs. 25% in people < 75 years) [65]. Importantly, data from the Framingham study demonstrated older age at stroke onset, but not gender or stroke type, to be associated with increased disability [66]. Further epidemiological studies revealed a strong age-dependency for the incidence of epilepsy [67], Parkinson’s disease [68], and dementia [69], with a continuous and strong growth in numbers in the elderly. Thus, the coincidence of CVDs and neurological diseases in the elderly is not surprising, due to shared risk factors, which themselves express a high age-dependency, such as hypertension, diabetes, vascular dysfunction, and atherosclerosis, accompanied by altered molecular mechanisms centered on inflammation and adverse redox signaling. 

## 4. The Oxidative Stress Concept of Aging

In 1954, Harman first described the “free radical theory of aging” [70]. He reasoned that since aging is a universal phenomenon, the underlying causation must also be universally present in all organisms. To this end, the focus shifted toward the hydroxyl radical and molecular oxygen being important mediators of the aging process [71]. Mitochondria are prolific producers of ROS within the cell, so they were natural targets for investigation within this theory. Since this high concentration of mitochondrial ROS (mtROS) is likely partly responsible for the high mutation rate of mtDNA, it is therefore necessary that two spatially separated genomes (nuclear and mitochondrial) co-exist, and both are required for the assembly of the respiratory chain components [72]. Further, as the mitochondrial genome malfunctions, irregularities in physiology and ATP synthesis are also seen, which are accompanied by amplified ROS generation and increased apoptosis [73]. Within the context of aging, the focus shifted away specifically from the hydroxyl radical and onto another free radical species, ^•^NO, which is now known to be an important vasodilator, to play a role in vascular smooth muscle cell proliferation, and to inhibit platelet aggregation, amongst other important regulatory roles. The age-dependent impairment of vascular redox balance is strongly linked to the bioavailability of the ^•^NO radical [74], which could be reduced through consumption by superoxide, and consequently lead to impaired vasorelaxation [36,75]. ^•^NO could thereby potentially serve as a biomarker for age-dependent endothelial dysfunction.

The free radical theory of aging was amended in 1972 by Harman to delineate the specific role of mitochondria [76], which were then moved to the forefront of the field. Harmon proposed that the mitochondrion was the primary origin of oxidative stress as well as the target—mitochondria produce a significant amount of cellular energy but are also damaged by ROS, which can attack both mitochondrial and nuclear DNA and can cause significant damage. With age, the damage accrued can result in defective mitochondria, which produce more and more ROS and in turn cause more oxidant-induced mutations and deletions, and culminate in a loss of cellular function, apoptosis, and necrosis. To this end, oxidant-induced damage in mtDNA was reported in the form of 8-oxo-deoxyguanosine (8-oxo-dG) [77,78], a mutagenic lesion whose accumulation was linked to pathological processes [79], and inversely correlated with lifespan of short-lived animals in the nuclear DNA and mtDNA of cardiac tissue. In brain tissue of long-lived animals, however, 8-oxo-dG content was higher in nuclear DNA (data not shown) [80]. These insights could be partially explained by higher metabolic rate, lower antioxidant clearance and defense, and possibly less efficient DNA repair. In this manner, genomic instability and cellular senescence occur as a result of age-related oxidative stress-induced DNA damages associated with shortened telomeres, increased DNA methylation, and decreased DNA content, all of which contribute to numerous degenerative and aging-related diseases [81]. In two mouse knockout models (*ALDH-2*^−/−^, *MnSOD*^−/−^), we found that mitochondrial ROS, mitochondrial DNA (mtDNA), and vascular dysfunction positively correlated with age [82]. Further, our data showed a correlation between endothelial dysfunction and mitochondrial ROS, which itself was mostly dependent on age, but secondarily dependent on the level of antioxidant enzymes present. Our data also showed a correlation between mtROS and mtDNA strand breaks, which led to a reasoning that mtDNA strand breaks arise from mtROS through direct interaction and oxidative DNA lesions, and given enough time and stress, could result in mitochondrial uncoupling and a secondary increase in ROS generation (through impaired de novo synthesis of functional respiratory complexes, due to mtDNA degradation or mutation). The ultimate message of the free radical hypothesis of aging is that ROS cause substantive alterations in biological macromolecules over the organism’s lifespan, which accumulate to detrimental effect [83]. The conclusion can be made that accumulation of DNA damage in sum cannot be a “one size fits all” predictor of total life years, but that the kinetics of formation and repair of DNA damage will vary, depending on species and tissue.

However, ROS generation is not the sole factor in the slow degradation of vascular function. Antioxidant defense and clearance are also impacted by age. For example, cytosolic superoxide dismutase 1 (SOD1) [75,84], mitochondrial superoxide dismutase 2 (SOD2) [85], extracellular SOD (ecSOD), and thioredoxin-1 (Trx) [86] showed both age- and expression-related reductions in clearance efficacy, as reflected by studies of endothelial function in young and old mice. If superoxide is the major contributor to vascular aging, the question arises—why are these antioxidant systems seemingly unable to defend against increasing levels of oxidative stress? To that point, in aging vessels, SOD2 was found to be heavily nitrated, and its activity thereby impaired. These findings were accompanied by increased 3-nitrotyrosine staining, which implies a role for peroxynitrite, a product of superoxide and ^•^NO as the nitrating agent [87]. It is obvious then that oxidative burden can cause inhibition of this protective enzyme, which then perpetuates a vicious cycle leaving the enzyme unable to perform effectively. Though it would be intuitive to expect that the overexpression of antioxidant enzymes would result in expansion of lifespan, this was not shown (*SOD2*^+/−^ or *SOD2*^tg^, *GPx-1*^−/−^, *GPx-4*^−/−^ or *MsrA*^−/−^, *SOD1*^tg^, *catalase*^tg^) [88], though overexpression of *Trx1* was shown to increase lifespan and stress resistance [89]. Conversely, only *SOD1*^−/−^ mice and mice with double gene ablation combinations reduced life expectancy [88,90], and *SOD2* knockouts do not survive past a few weeks from birth [91,92]. Taken in conjunction, these data suggest that antioxidant systems are critically important to life, but also that there is a “cap” to their beneficial effects. This further implies that it is not the absolute amount of oxidative stress that impacts lifespan, but rather, a balance that must be maintained. While oxidative stress might not be the direct determinant in aging, as previously hypothesized [88,90,93], the contribution of oxidative stress in aging seems to be a factor that prevents healthy aging by impacting organ function [94,95,96,97].

The length of time an organism remains healthy is another factor through which antioxidant enzymes could have a significant effect and a notable clinical importance. This “healthspan” could be indicated by the lack or decreased progression of age-related cardiovascular complications and the resistance to stress conditions during normal aging [89]. In studies utilizing genetic deletion of aldehyde dehydrogenase-2 (*ALDH-2*) and manganese superoxide dismutase (*MnSOD*), two important mitochondrial antioxidant proteins, we found that mitochondrial oxidative stress and vascular dysfunction arose as a function of aging [82], which supports the idea that mtROS is especially important in the degree of health in aging [94,95,96,97].

Aside from mitochondrial ROS, there are other cellular sources of ROS that have an impact on the healthspan. The state of eNOS plays an important role in whether it produces a vascular hero, ^•^NO, or a villain, O_2_^•−^ [98]. In the coupled state, eNOS consists of a protein dimer and two BH4 cofactors that facilitate electron transfer needed for L-arginine oxidation and production of ^•^NO [54,55,56]. When BH4 is either oxidized to BH2 or absent, or electron flow from the reductase to the oxygenase domain is impaired by either eNOS S-glutathionylation or adverse phosphorylation, the eNOS dimer is uncoupled and produces ROS in the form of O_2_^•−^ (which is why NOS enzymes are also called Janus-faced enzymes). The overproduction of O_2_^•−^ further oxidizes BH4 and inhibits ^•^NO synthesis. The result of eNOS uncoupling is then a reduction in ^•^NO bioavailability [99] and can contribute to the pathogenesis of endothelial dysfunction in aged vessels [9,36]. It was reported from several sources that eNOS expression levels rise with age, which could possibly be to counteract the effect of eNOS uncoupling and reduced ^•^NO bioavailability. There are also groups who report unchanged eNOS expression in aged vessels, but instead report decreases in Akt-dependent phosphorylation of eNOS^Ser1177^. Either reports could be consistent with the findings of endothelial dysfunction in aged vessels and elderly individuals [100]. We additionally reported both S-glutathionylation by PKC and adverse phosphorylation of eNOS at Thr495 and Tyr657 by PYK-2, as important redox-sensitive mechanisms in the process of aging-induced vascular dysfunction [101].

## 5. Oxidative Stress and Inflammation by Air Pollution

The modern world functions on the movement of people and goods, both of which require the consumption of fuel. The byproducts of this consumption, diesel exhaust, particulate matter (PM), and ultrafine particles, were demonstrated to be a contributing factor in the pathogenesis and progression of CVD (Figure 2) [15,102]. Early studies in the field implicated these particles in the development of endothelial dysfunction in rat aortas exposed to diesel exhaust particulate, effects that were mitigated by pre-treatment with superoxide dismutase [103]. Exposure studies in animals revealed that in the setting of atherosclerosis, plaque development was accelerated [104,105]. Body scans by positron emission tomography in men revealed that particulate matter induces coronary inflammation, which was associated with higher cardiovascular risk [106]. In further human studies, platelet hyperreactivity and thrombus formation were observed [107,108]. The prevailing hypothesis for how inhaled PM causes cardiovascular effects is through immune cell activation in the lung. Activated immune cells release cytokines into circulation, causing an inflammatory cascade generating ROS through NADPH oxidase in activated immune cells and the vascular wall, consequently decreasing ^•^NO bioavailability. Diesel exhaust particles also possess active surfaces that are capable of directly stimulating the formation of superoxide, representing a second possible manner through which ROS could be produced (as demonstrated by TEMPO spin trapping) [103]. It is also possible that inhaled particles mechanically stimulate the alveolar sensory receptors, leading to effects on the cardiovascular system via the autonomic nervous system. These two mechanisms could possibly work in a synergistic way to impart the effects. An overview focusing on the cardiovascular oxidative stress pathways through which PM might interact is provided by Rao et al. [109].

Investigations on PM health effects seeking mechanistic insight takes place in both animals and humans. Mechanistic studies in animals show clear endothelial dysfunction, leukocyte activation, and progression of atherosclerotic plaques [105]. These results point toward oxidative stress carrying an important role in the onset of these effects, as they are shared by other more “traditional” risk factors, such as diabetes, hypertension, and smoking [110]. Underlining this point, studies in mice with a p47phox global knockout were largely protected from the metabolic effects of PM exposure (insulin resistance) and showed improved vascular function and visceral inflammation [111]. Additionally, microvessels from mice treated with metal-rich PM_2.5_ showed normalized endothelial function upon ex vivo incubation with apocynin or VAS2870, both NADPH oxidase inhibitors [112]. Additionally, eNOS uncoupling was prevented by apocynin in cultured endothelial cells upon incubation with ultrafine particles [113]. Though PM exposure leads to mitochondrial dysfunction and mtROS formation, research into the impact of these factors on the phenotype arising from PM exposure is still needed. However, not all experimental data support this concept. In a small cohort trial, 18 healthy subjects were exposed for 3 h to diesel exhaust at 276 g/m^3^ from a passenger car or filtered air, with co-exposure to traffic noise at 48 or 75 dB(A) [114]. Exposure to diesel exhaust had no effects on genotoxicity, oxidative stress (DNA single strand breaks and 8-oxo-dG lesions measured by formamidopyrimidine (fapy) DNA glycosylase-induced artificial strand breaks), or inflammation in white blood cells isolated from the subjects, whereas exposure to noise caused oxidative DNA damage [115].

There is an obvious inflammatory involvement in PM-induced cardiovascular consequence that likely contributes to ROS generation. PM can present as a “danger signal” or damage-associated molecular patterns (DAMP) via CD44, TLR4, and CD36 to trigger activation of NFκB, kinase signaling, and cytokine synthesis [116]. Downstream, TLR4 is a direct activator of NOX2, an important producer of superoxide in phagocytes [117]. A cohort study of 18 highway maintenance workers showed that their exposure to PM and traffic noise were associated with C-reactive protein, serum amyloid A, increased heart rate variability, or systolic, and diastolic blood pressure [118]. Overall, exposure to PM is likely to work through a variety of oxidative and inflammatory mechanisms, to cause serious cardiovascular complications, which seem to be exacerbated in the setting of established disease and other risk factors, including noise.

Two additional pathways might be important as well. Sirtuins might play a role in air pollution-induced disease and aging, as they are important regulators of antioxidant defense via control of the acetylation status of NRF2 and FOXO3a [119,120]. A study in mice showed that sirtuin1 protects against PM_2.5_-induced lung coagulation and inflammation, by exposing *sirtuin1* knock-out mice, who were highly susceptible to these disorders [121]. Analysis of the Chinese Longitudinal Healthy Longevity Survey with 7083 participants, revealed that there is an increase in the mortality hazard ratio corresponding to certain sirtuin1 alleles (1336 vs. 1078 for participants carrying two *SIRT1_391* minor alleles vs. participants carrying one or none minor alleles) [122]. The mammalian target of rapamycin (mTOR) inhibits autophagy, a natural and essential biological process through which cells recycle damaged and dysfunctional organelles [123]. Therefore, blocking of the mTOR pathway promotes longevity but might also cause impairment of energy metabolism and induce long-term health consequences [124]. In mice, PM_2.5_ exposure causes overexpression of mTOR, leading to apoptosis [125], as well as pulmonary inflammation and fibrosis [126].

## 6. Oxidative Stress and Inflammation by Traffic Noise Exposure 

Noise is another newly studied cardiovascular risk factor with strong ties to oxidative stress pathways. Noise appears to exert its effects through stress mechanisms, which subsequently cause a cascade of oxidative, inflammatory, and metabolic effects that work in tandem to cause vascular damage and lead to higher cardiovascular risk (Figure 2). This noise stress concept was established by Babisch [127,128] and was recently confirmed at the molecular level, through positron emission tomography scans in men demonstrating amygdala activation due to severe noise exposure, which was associated with more pronounced coronary atherosclerotic/inflammatory alterations and higher cardiovascular risk (more major adverse cardiovascular events) [129]. Other studies support this mechanistic line, including a study in rats exposed to octave band noise for 8 h a day over 20 days (80–100 dB(A), 8–16 kHz, 8 rats/group). Increased levels of stress hormones corticosterone, adrenaline, and noradrenaline, as well as vasoconstrictor endothelin-1 were found in the plasma of these rats [130], supporting the hypothesis that stress is induced by noise exposure. The effects of stress hormones on the physiology are diverse but are generally associated with increased blood pressure and heart rate [131]. Effects on endothelium-dependent vasodilation were also seen in rats exposed to 100 dB(A) of noise for 2 and 4 weeks. Rings from the thoracic aorta showed blunted response to acetylcholine and increased sensitivity to serotonin, but not to phenylephrine or potassium chloride, clearly showing an endothelium-mediated effect in vasoconstriction. Furthermore, an increase in systolic blood pressure of 31 mmHg was also recorded [132]. Studies in rats also demonstrated higher levels of ROS in the cerebral cortex, alongside reduced levels of ^•^NO in the cerebellum and brainstem, which was reduced with rosuvastatin, a cholesterol-lowering drug with antioxidative effects [133]. Taken in sum, these indicate that the effects of noise are not specific to the vasculature but might also have effects in systems other than the cardiovascular.

Beyond studies in rodents, field studies in humans were also conducted with regard to noise exposure. One such study of overnight at-home noise exposure to aircraft noise yielded results that showed significant endothelial dysfunction, as measured by flow-mediated dilation of the brachial artery [134]. Acute administration of vitamin C rescued the endothelial function of these study participants, very clearly indicating either insufficient ROS clearance or overactive ROS generation as the culprit. Repeated exposure to noise can “prime” the vasculature for damage, where over time, the effects of oxidative stress overwhelm antioxidant defense and incur lasting damage. Echoing the results seen in rodents, study participants also had increases in catecholamine production, indicating stress responses that can lay the foundation to CVD development. The HYENA study, a field study of 4861 people living near major airports in Europe, was unable to associate daytime noise exposure with increases in blood pressure, but did find significant increases in blood pressure in participants exposed to nighttime noise exposure [135]. Last, we found endothelial dysfunction, sleep disruption, and an increase in blood pressure to be exacerbated in patients with coronary artery disease [136]. There was no correlation between those who responded to be annoyed by or sensitive to noise, implying that the effects were not via an emotional pathway and that noise-induced damage would occur independent of cognition. Conversely, population-based cohort studies clearly demonstrated a dose-dependent increase of prevalent atrial fibrillation [137], accompanied by increased midregional pro-atrial natriuretic peptide levels [138], a cardiac hormone that mirrored endothelial activation and predicted future cardiovascular events, in response to annoyance due to different noise sources, including traffic noise. These effects could possibly be impactful on future cardiovascular events [16,139].

We established a protocol for aircraft noise exposure consisting of 69 43-s-long aircraft noise events, irregularly spaced over 2 h, interspersed with silent periods, in order to ascertain the effects of noise exposure on the vasculature, in a systemic manner (maximum sound pressure level (SPL) of 83 dB(A), mean 71.6 dB(A), 50–55 dB(A) in silent periods) [140]. The noise events were repeated constantly over the course of 1, 2, and 4 days. We observed increases in catecholamines, angiotensin-II, and endothelin-1 levels in plasma or systolic/diastolic blood pressure, as well as impaired ^•^NO signaling, supersensitivity to vasoconstrictors, and endothelial dysfunction over the entire duration of the noise exposure. Noise-exposed animals had increases in oxidative stress markers, such as eNOS uncoupling (L-NAME-sensitive superoxide signal and eNOS S-glutathionylation), 3-nitrotyrosine- and malondialdehyde-positive proteins, endothelin-1 and NOX-2 protein expression, accompanied by increased dihydroethidium signal (marker of ROS production) and immune cell infiltration in the vascular wall. None of these effects were replicated when mice were exposed to white noise [140]. In a subsequent study, we could also demonstrate neuroinflammation, cerebral oxidative stress, and circadian dysregulation, upon aircraft noise exposure and prevention of all adverse effects in noise-exposed *Nox2* knockout mice [141]. Figure 3 shows the pathomechanisms for increasing the risk of cardiometabolic disease, induced by air pollution and noise exposure. The impact of noise on sirtuin and mTOR pathways is not well studied (only few reports on models of hearing loss are available, which is not within the focus of the present review).

## 7. Human Evidence on the Association between air Pollution and Biomarkers of Aging

There is ample evidence from human studies suggesting that exposure to multiple air pollutants associates with increased biological aging, mainly examined on the basis of telomere length, mtDNA content, and DNA methylation. In the studies presented here, telomere length was basically used as a biomarker of the aging (senescence) process. Pieters et al. investigated the association between long-term exposure to PM with telomere length and mtDNA content, in a Belgium sample of 166 non-smoking elderly subjects [143]. After multivariable adjustment, the authors found an increase in PM_2.5_ levels per 5 μg/m^3^ to be associated with a relative decrease of 16.8% (95% CI −26.0 to −7.4) in telomere length and a relative decrease of 25.7% (95% CI −35.2 to −16.2) in mtDNA content. Likewise, a study of high PM-exposed workers (N = 240) from China revealed an association between long-term PM_10_ exposure and telomere shortening (−9.9%, 95% CI −17.6 to −1.5 per IQR increase) [144]. Data from the NAS cohort demonstrated a 7.6% decrease (95% CI −12.8 to −2.1 per IQR increase) in telomere length, in response to long-term black carbon exposure in 165 never-smoked elderly men, with subgroup analysis indicating stronger effects for older (≥75 years of age) than younger subjects [145]. A meta-analysis of three cohort studies from Germany and the USA by Panni et al. indicated substantial associations between DNA methylation (cytosine-guanine dinucleotide sites) and short- and mid-term PM_2.5_ exposure [146]. In the German KORA cohort study (N = 1777), associations between multiple air pollutants and various biomarkers of aging including telomere length and other epigenetic measures were assessed in a sex-specific manner [147]. A multiple phenotype analysis combining all aging measures revealed long-term exposure to black carbon and PM_10_ to be broadly associated with biological aging in men. In a U.S. cohort of 2747 women from the Sisters Study, long-term exposure to NO_2_ was inversely associated with DNA methylation age acceleration (β = −0.24, 95% CI −0.47 to −0.02 per IQR increase), whereas no association was observed for PM_10_ [148].

The relationship between air pollution and telomere length in adults was the subject of a recent systematic review and meta-analysis from Miri et al., including 19 observational studies (11 retrospective and seven prospective studies) [149]. Herein, long-term PM_2.5_ exposure was inversely related to telomere length, as indicated by the meta-analysis of two studies (−0.03, 95% CI −0.05 to −0.01 per 5 μg/m^3^ increase). The meta-analysis of three short-term PM_2.5_ studies (0.03, 95% CI 0.02 to 0.04) and two polychlorinated biphenyls studies (0.10, 95% CI 0.06 to 0.15) showed positive associations with telomere length. In contrast, no associations were found between short-term exposure to PM_10_ and polycyclic aromatic hydrocarbon exposure and telomere length, respectively. In line with these results, it is widely assumed that short-term exposure to air pollutants is related to increased telomere length, in the manner of an acute adaptive response, whereas long-term exposure with subsequent cumulative burden of oxidative stress and inflammation is associated with shorter telomere length. Conflictingly, in a prospective U.S. cohort of 772 critically ill patient’s long-term exposure to ozone was associated with increased telomere length after adjustment for potential confounders, which remained stable for the analysis of subgroups with sepsis, trauma, and acute respiratory distress syndrome [150]. Further analyses of air pollutants including PM_2.5_, PM_10_, CO, NO_2_, and SO_2_ revealed no associations with telomere length, whereas higher warm-season PM_2.5_ and CO exposures were independently associated with increased telomere length. A wide range of studies investigated the relationship between prenatal exposure to air pollution and biological aging at birth (for review see [151]). In this context, Martens et al. examined the association of prenatal exposure to PM and newborn telomere length in a prospective birth cohort of 730 mother–newborn pairs from Belgium. After multivariable adjustment, an inverse relationship was found between PM_2.5_ exposure (per 5 μg/m^3^ increase) during the entire pregnancy, and cord blood leukocyte telomeres (−8.8%, 95% CI −14.1 to −3.1) and placental telomere length (−13.2%, 95% CI −19.3 to −6.7) [152]. A study on 200 Iranian preschool children (5–7 years old) found exposure to higher levels of ambient PM_1_, PM_2.5_, and PM_10_ at home and kindergartens to be associated with a shorter telomere length [153].

## 8. Human Evidence on the Association between (Traffic) Noise Exposure and Biomarkers of Aging

Since inflammatory and oxidative stress processes are fundamentally involved in the pathogenesis of noise-induced health conditions, along with the fact that noise and air pollution usually co-exist [16,23], it is of special importance to determine the impact of noise exposure on measures suggestive of biological aging. Recent studies primarily focused on inflammatory and oxidative stress pathways in response to traffic noise exposure in the context of CVD development (for review see [22,23,131]), while human data linking noise exposure to measures such as telomere length and DNA methylation are generally lacking. Evidence on the association between traffic noise exposure and biological aging mainly arises from animal studies [154,155,156,157,158,159]. Indirect evidence on the role of noise exposure in biological aging is given by human studies demonstrating oxidative DNA damage, a correlate of altered DNA methylation and gene expression, in subjects exposed to occupational [160,161] and traffic noise [114]. However, there are only few human studies suggesting that noise exposure, like air pollution, might also influence health via DNA methylation and telomere length.

Eze et al. were first to examine the association between long-term exposure to traffic noise, air pollution, and DNA methylation in 1389 Swiss subjects from the SAPALDIA study [162]. DNA methylation was independently associated with measures of transportation noise (aircraft, railway, and road traffic noise) and air pollution (NO_2_ and PM_2.5_), with enrichment for pathways related to inflammation and immune response, which might explain at least in part the relationship between these exposures and various age-related outcomes, as the authors concluded. In the NESDA cohort (N = 2902), neighborhood quality comprising measures of perceived neighborhood disorder, fear of crime, and noise (“how often do you experience noise from neighbors, traffic or other sources in your neighborhood?”) was determined to assess its relation with telomere length, after comprehensive adjustment for individual and community level variables [163]. The results revealed a 69 base pair shorter mean telomere length in subjects who reported moderate neighborhood quality (β −69.33, 95% CI −119.49 to −19.17) and a 174 base pair shorter mean telomere length in subjects who reported poor neighborhood quality (β −173.80, 95% CI −298.80 to −49.01), compared to subjects who reported good neighborhood quality. Importantly, these outcomes corresponded to 8.7 and 11.9 years in chronological age, respectively. However, when looking specifically at the subdomain of noise, no associations were observed with telomere length after multivariable adjustment. Of note, in these studies telomere length was basically used as a biomarker of the aging (senescence) process.

Since chronic exposure to excessive noise levels is among the leading risk factors for hearing loss mediated by noise-induced inflammation, oxidative stress, and endothelial damage, further evidence of a contribution of noise in biological aging might arise from studies examining the association between biomarkers of aging and hearing loss. A recent case-control study from China examined the association between telomere length and risk of hearing loss, including 817 cases and 817 matched control subjects [164]. Decreased odds of hearing loss was observed for subjects in the highest quartile of telomere length, compared to the lowest quartile (OR 0.53, 95% CI 0.38 to 0.74). In good agreement, a further case-control study from China demonstrated decreased odds of hearing impairment with increasing telomere length [165].

## 9. Conclusions

Despite apparent differences in symptomology, CVDs and neurological diseases have a clear overlap in risk factors, mechanisms, and possibly etiology, which likely all work in concert. Most CVD and metabolic disease such as diabetes as well as neurological diseases have associations with inflammation and oxidative stress, which seemingly suggests that any factor that impacts those could potentially have great clinical significance. Aging, air pollution, and noise exposure all have strong ties connecting these risk factors with oxidative stress, mtROS or otherwise, and inflammation. While the mechanisms truly underlying the onset of aging are not fully elucidated, there is clear evidence for a convergence with other risk factors, including environmental stressors, at the level of oxidative stress and inflammation. Noise induces a hormonal stress response via the HPA axis, which then triggers a cascade of cytokine production, immune cell migration, and ROS production. Air pollution activates very similar pathophysiological pathways but, in addition, can cause direct damage at the level of blood vessels and organs (either by interaction of small particles with phagocyting immune cells such as neutrophils and macrophages, or by direct ROS formation on the active surfaces of the particles). In the elderly, low-grade inflammation is known to contribute to a higher prevalence of metabolic and cardiovascular complications [166]. Our data [140] and reports by others [118] also clearly show that air and noise pollution induce an inflammatory “kickstart”, and in the presence of other risk factors, even exacerbate the underlying pathophysiology, as seen in patients with coronary artery disease [136], and hypertensive mice with noise exposure [167]. Figure 4 displays an overview of the pathomechanisms, underlying the aging process, as well as the environmental risk factors of air pollution and traffic noise exposure, to increase the risk of CVD and neurological disease.

While there are no studies specifically examining these environmental risks in the elderly, it is conceivable that these factors could exacerbate an underlying immunosenescence that either accompanies or produces age-related disease and common comorbidities [4,5,6]. Data from a recent study in an animal model of metformin-dependent AMP-activated protein kinase (AMPK) activation, demonstrated an increased lifespan and healthspan in elderly animals, due to the improvement of inflammatory and oxidative damage [168]. Normalizing the effects of inflammation and equally importantly, oxidative stress, could be a promising strategy for improving healthspan [169]. Since mtROS formation increases with age [34] and activates immune cells along with their NADPH oxidase, thereby stimulating cytokine release and the inflammasome [170,171,172,173,174,175], mtROS is responsible for initiating a self-perpetuating cycle of oxidative stress and inflammation in the aged. Coupled with additional risk factors, including noise and air pollution, intervention by quenching mtROS production seems an elegant method of promoting healthy aging. As these stressors have a point of convergence with aging processes, there is potential to affect all of them through a single interventional route such as antioxidant pharmacological (e.g., dietary AMPK [176] or NRF2 [177] activation) and non-pharmacological interventions (e.g., physical exercise [178,179] and intermittent fasting), possibly making huge impacts in epidemiological outcomes. However, while the exact mechanisms through which environmental stressors accelerate aging and contribute to increased risk of age-related diseases as well as specific treatment options, remain to be established, there is general consensus that we urgently need measures to reduce exposure to environmental pollutants. In the setting of traffic noise, a variety of mitigation strategies are proposed, such as developing and using low-noise tires, quiet engines, and breaks, applying quiet surfaces, installing noise barriers in densely populated areas, introducing adequate speed limits (in particular during sleeping hours), minimizing the overlap of traffic routes and housing zones, introducing night bans, installing sound-reducing windows, and placing sleeping rooms towards the quiet side of the house. Current approaches to mitigate air pollution exposure are mainly focused on personal actions like using air filtration or face masks, avoiding exposure by changing travel routes, staying indoors/closing windows, modification of outdoor activities, and keeping distance from areas where higher concentrations of air pollutants are expected, such as major roadways [180]. In any case, large-scale macro interventions in the long run, on a political and societal level that question the way we work and live are clearly needed to achieve sustainable success.

## Figures and Tables

**Figure 1 ijms-22-02419-f001:**
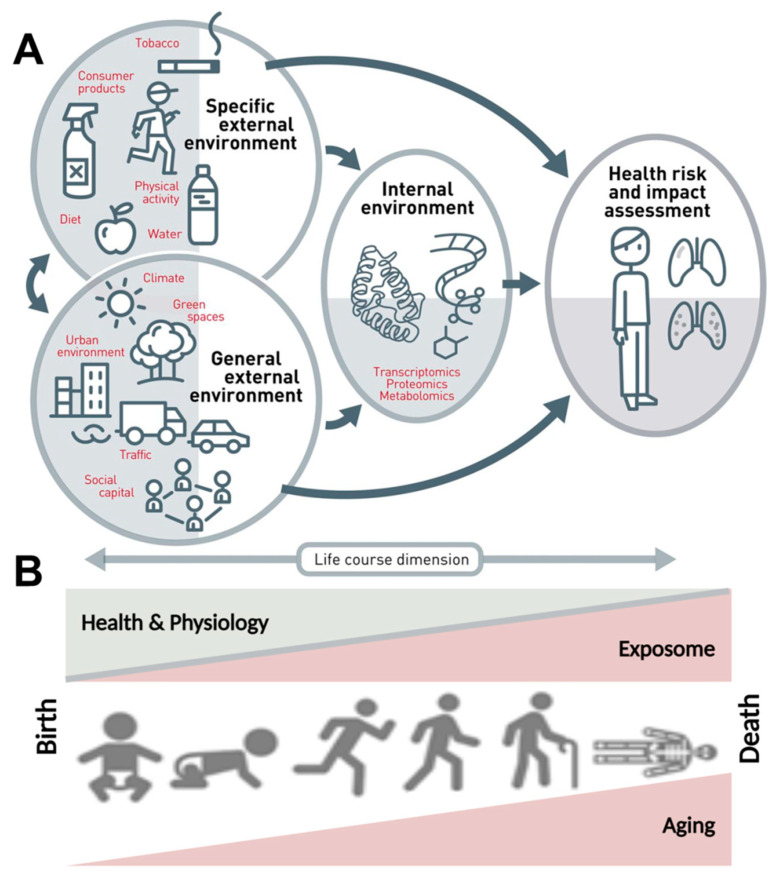
The exposome concept. (**A**) The exposome comprises the totality of a person’s external and internal exposures, from birth to death. (**B**) The external exposures and their internal exposure-related biochemical changes accumulate steadily over the aging process and lead to altered health risks. Adapted from Vrijheid et al. [2] (upper part, Copyright © 2021, BMJ Publishing Group Ltd. and the British Thoracic Society) and Misra [3] (lower part, under the terms of the Creative Commons Attribution License (CC BY), Copyright © 2021 Misra) with permission.

**Figure 2 ijms-22-02419-f002:**
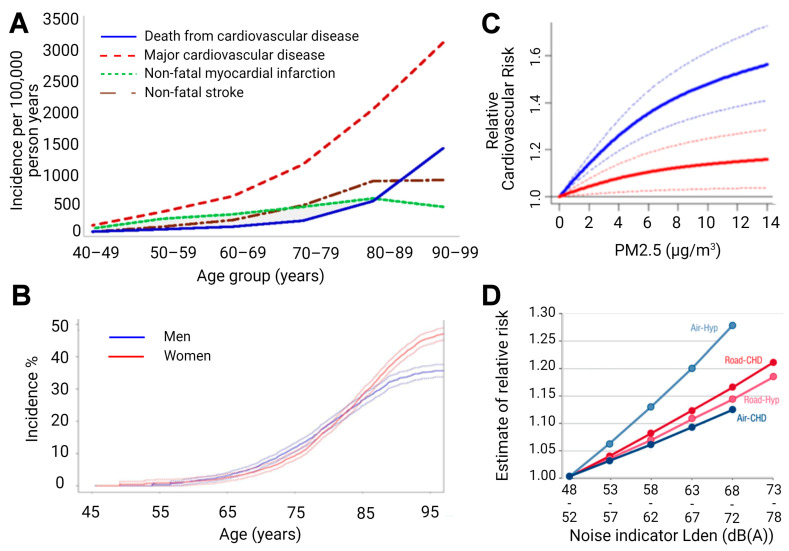
(**A**) Age-specific crude incidence of confirmed major cardiovascular disease by type of first event (non-fatal myocardial infarction, non-fatal stroke, and death from cardiovascular disease). Reused from [13] with permission under the terms of the Creative Commons Attribution Non-commercial License, Copyright © Driver et al. 2008. (**B**) Risk of common neurological diseases for 45-year-old men and women. In this analysis, follow-up ended at time of first occurrence of dementia, stroke, or parkinsonism. For instance, for individuals who first suffered a stroke and subsequently developed dementia, only the stroke event is considered. Reused from [14] with permission, Copyright © 2021, BMJ Publishing Group Ltd. All rights reserved. (**C**) Predicted values of relative risk for cardiovascular mortality by chronic exposure to increasing particulate matter concentrations for high ozone levels (37.60 ppb, solid blue line) and low ozone levels (20.26 ppb, solid red line) with uncertainty intervals (dashed lines). Reused from [15] with permission under the terms of the Creative Commons CC BY license, Copyright © 2021, The Author(s). (**D**) Exposure-response relationships for the associations between transportation noise and cardiovascular health outcomes. Road—road traffic noise, Air—aircraft noise, Hyp—hypertension, CHD—coronary heart disease, and Lden—day-evening-night level, i.e., the average sound pressure level measured over a 24-h period. Reused from [16] with permission, Copyright © 2021, Oxford University Press.

**Figure 3 ijms-22-02419-f003:**
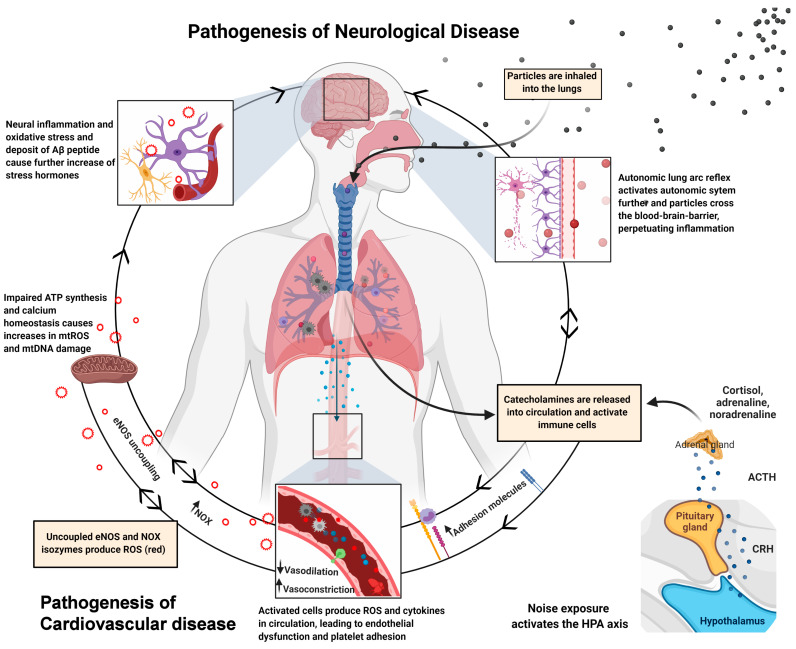
Pathophysiological mechanisms of neurological and cardiovascular disease induced by air pollution and noise exposure. Arrows indicate directions of pathways and pathomechanisms. Red circles indicate the release adverse signaling ROS. Summarized from Münzel et al. [142] and presented data in the present review article. Created with BioRender.com.

**Figure 4 ijms-22-02419-f004:**
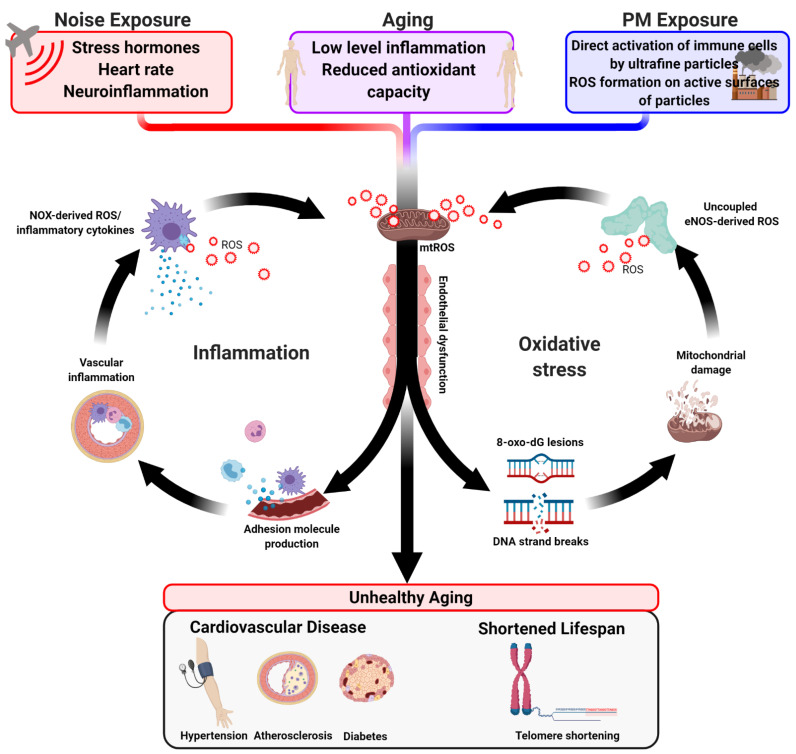
Convergence of pathomechanisms underlying the aging process as well as the environmental stressor traffic noise exposure and air pollution, leading to an increased risk of CVD and neurological disease. Blue dots indicate released adverse signaling cytokines and red circles the release of detrimental ROS. Created with BioRender.com.

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
