# Peer review of "Accelerated Aging and Age-Related Diseases (CVD and Neurological) Due to Air Pollution and Traffic Noise Exposure"

_ijms, 2021, doi:10.3390/ijms22052419_

Round 1
Reviewer 1 Report
The manuscript is certainly a good report on the published data on the role of environmental factors on aging processes and it will be of great use for scientists involved in the field. It is not always clear the connection between some biological markers ( for example : telomere shortening) and the development of the diseases. The conclusion should have expanded more on the possible intervention studies to reduce the effects of environmental stressors.Author Response
The manuscript is certainly a good report on the published data on the role of environmental factors on aging processes and it will be of great use for scientists involved in the field. It is not always clear the connection between some biological markers ( for example : telomere shortening) and the development of the diseases. The conclusion should have expanded more on the possible intervention studies to reduce the effects of environmental stressors.
Response: We thank you for the highly favorable evaluation of our work. Telomere shortening was rather used as a biomarker of the aging (senescence) process here, and rather not as a trigger of disease. In line with the first comment, we have made this clear in section 7 and 8 by adding respective notions.
“In the here presented studies, telomere length was basically used as a biomarker of the aging (senescence) process.”
In addition, we have rearranged and clarified the section on mitochondrial and nuclear damage arising from ROS, to be more succinct and direct in its manner of describing pathogenesis and to make the link clearer between biological markers such as telomere shortening and disease risk.
“Harmon proposed that the mitochondrion was the primary origin of oxidative stress as well as the target: mitochondria produce a significant amount of cellular energy but are also damaged by ROS, which can attack both mitochondrial and nuclear DNA and cause significant damage. With age, the damage accrued can result in defective mitochondria, which produce more and more ROS and in turn cause more oxidant-induced mutations and deletions, and culminate in a loss of cellular function, apoptosis, and necrosis. To this end, oxidant-induced damage in mtDNA has been reported in the form of 8-oxo-deoxyguanosine (8-oxo-dG) [77,78] a mutagenic lesion whose accumulation has been linked to pathological processes and inversely correlated with lifespan of short-lived animals in the nuclear DNA and mtDNA of cardiac tissue. In brain tissue of long-lived animals, however, 8-oxo-dG content was higher in nuclear DNA (data not shown) [79]. These insights could be partially explained by higher metabolic rate, lower antioxidant clearance and defense, and possibly less efficient DNA repair. In this manner genomic instability and cellular senescence occur as a result of age-related oxidative stress-induced DNA damages associated with shortened telomeres, increased DNA methylation, and decreased DNA content, all of which contribute to numerous degenerative and aging-related diseases [80].”
We further added intervention strategies to limit the exposure to air and noise pollution.
“However, while the exact mechanisms by which environmental stressors accelerate aging and contribute to increased risk of age-related diseases as well as specific treatment options remain to be established, there is general consensus that we urgently need measures to reduce the exposure to environmental pollutants. In the setting of traffic noise, a variety of mitigation strategies have been proposed such as developing and using low-noise tires, quiet engines, and breaks, applying quiet surfaces, installing noise barriers in densely populated areas, introducing adequate speed limits (in particular during sleeping hours), minimizing overlap of traffic routes and housing zones, introducing night bans, installing sound-reducing windows, and placing sleeping rooms towards the quiet side of the house. Current approaches to mitigate air pollution exposure are mainly focused on personal actions such as using air filtration or face masks, avoiding exposure by changing travel routes, staying indoors/closing windows, modification of outdoor activities, and keeping distance from areas where higher concentrations of air pollutants are expected such as major roadways [179]. In any case, large scale macro interventions in the long run on a political and societal level that question the way we work and live are clearly needed to achieve sustainable success.”
Reviewer 2 Report
This is a clear, concise and well-written review. I don't have any specific comments to give.
Some minor revisions can be made to improve the quality of its format, such as to enlarge the font size of Fig. 2B legend.
Author Response
This is a clear, concise and well-written review. I don't have any specific comments to give.
Some minor revisions can be made to improve the quality of its format, such as to enlarge the font size of Fig. 2B legend.
Response: Thank you for this positive feedback. In line with your suggestion, we improved the quality of all panels in figure 2 and also increased the font size of axis labels and legends.
Reviewer 3 Report
The review is very interesting, comprehensive and well written. I have some suggestions made in order to help the readers get through this complex matter.
The sentence; line 32: „In contrast to the primary health concerns of the early 20th century and prior, age-related ….” is confusing and complicated – please rephrase.
Line 44: “Cardiovascular disease (CVD) is foremost amongst these age-related diseases to have 44
seen an increase in prevalence (Figure 2), which creates an urgent need for research in the 45
field” – please rephrase.
Figure 2 panel B is unreadable. Enlargement of the font and adding better labeling is suggested.
I suggest to adjust the graphics of all panels in Figure 2. Now each panel uses different labeling fonts and style.
Line 130 “It was found that enhanced H3K27ac and H3K4me3” – please explain what are these histone modifications. If you refer to epigenetic changes it would be beneficial to add “Epigenetics” to figure 1 (upper panel).
Line 288 “important in in” please correct
Line 292 “When uncoupled, eNOS produces…” - explain the idea of eNOS uncoupling
Author Response
The review is very interesting, comprehensive and well written. I have some suggestions made in order to help the readers get through this complex matter.
Response: We thank you for the favorable comment on our manuscript.
The sentence; line 32: „In contrast to the primary health concerns of the early 20th century and prior, age-related ….” is confusing and complicated – please rephrase.
Response: We rephrased the sentence accordingly. “In contrast to maternal, infectious diseases that were widely prevalent and a primary health concern of the early 20th century, age-related diseases are often chronic and require continual treatment over an extended period of time, thus correlating increased lifespan with chronic disease onset and elevated expense burden.”
Line 44: “Cardiovascular disease (CVD) is foremost amongst these age-related diseases to have seen an increase in prevalence (Figure 2), which creates an urgent need for research in the field” – please rephrase.
Response: We rephrased the sentence accordingly. “The aging population is particularly susceptible to cardiovascular disease (CVD), demonstrating the leading cause of death in populations aged over 65 years (Figure 2) and creating an urgent need for research in the field.”
Figure 2 panel B is unreadable. Enlargement of the font and adding better labeling is suggested.
I suggest to adjust the graphics of all panels in Figure 2. Now each panel uses different labeling fonts and style.
Response: Thank you for this comment. We improved the quality of all panels in figure 2 and also increased the font size of axis labels and legends.
Line 130 “It was found that enhanced H3K27ac and H3K4me3” – please explain what are these histone modifications. If you refer to epigenetic changes it would be beneficial to add “Epigenetics” to figure 1 (upper panel).
Response: We have exchanged the abbreviations for their full names in order to make it clear which histone modifications are responsible for the observed effect and we added the explanation for the function of these histone modifications.
“It was found that enhanced histone 3 lysine 27 acetylation and histone 3 lysine 4 trimethylation (epigenetic markers usually associated with transcriptional activation) promoted the transcriptional activation of cardioprotective genes such as eNOS, SOD2, GPx-1, and HO-1 which explained the observed heritable effects.”
We acknowledge the comment of the reviewer to add “Epigenetics” to Figure 1, but we think that changing the figure used from a paper of other colleagues will not be allowed within the permission terms. Also, the term “Transcriptome” also comprises epigenetic mechanisms such as microRNA signaling.
Line 288 “important in in” please correct
Response: We corrected the sentence.
Line 292 “When uncoupled, eNOS produces…” - explain the idea of eNOS uncoupling
Response: We have modified the sentences in lines 292-297 to better explain the idea of eNOS uncoupling.
“Aside from mitochondrial ROS, there are other cellular sources of ROS which make have an impact on the healthspan. The state of eNOS plays an important role in whether it produces a vascular hero, •NO, or a villain, O2•‾ [98]. In the coupled state, eNOS consists of a protein dimer and two BH4 cofactors that facilitate electron transfer needed for L-arginine oxidation and production of •NO [54-56]. When BH4 is either oxidized to BH2 or absent, or electron flow from the reductase to the oxygenase domain is impaired by either eNOS S-glutathionylation or adverse phosphorylation, the eNOS dimer is uncoupled and produces ROS in the form of O2•‾ (that is why NOS enzymes are also called Janus-faced enzymes). The overproduction of O2•‾ further oxidizes BH4 and inhibits •NO synthesis. When uncoupled, eNOS produces ROS, which in turn oxidize BH4, an im-portant prerequisite in •NO synthesis. The result of eNOS uncoupling is then a reduction in •NO bioavailability [99] and can contribute to the pathogenesis of endothelial dys-function in aged vessels [9,36]”.